# Supercontinuum Generation in the Cladding Modes of an Endlessly Single-Mode Fiber

**Tobias Baselt** [1,2,3,*] **, Bryan Nelsen** [1,2] **, Andrés Fabián Lasagni** [2,3] **and Peter Hartmann** [1,2]

1    Faculty of Physical Engineering/Computer Sciences, University of Applied Sciences Zwickau,
     Zwickau D-08056, Germany; bryan.nelsen@fh-zwickau.de (B.N.); peter.hartmann@fh-zwickau.de (P.H.)
2    Fraunhofer Application Center for Optical Metrology and Surface Technologies AZOM, Fraunhofer Institute
     for Material and Beam Technology IWS, Dresden D-01277, Germany; andres_fabian.lasagni@tu-dresden.de
3    Faculty of Mechanical Science and Engineering, Institute of Manufacturing Technology,
     Technische Universität Dresden, Dresden D-01062, Germany
*    Correspondence: tobias.baselt@fh-zwickau.de



**Featured Application: Supercontinuum generation in the cladding of a photonic crystal fiber was implemented to overcome the damage threshold that occurs during the core-pumped scenario by spreading the pump over a much larger area. The conversion efficiency in the visible tail of the spectrum is increased over that of the core-pumped case, which suggests phase matching in the quasi-continuum of cladding modes.**

**Abstract:** In photonic crystal fibers, light guidance can be achieved by a central defect of a periodic structure of air holes in a silica glass matrix and the dispersion can be adjusted over a wide spectral range to enhance nonlinear effects. By coupling short pulse laser radiation into the core with tight confinement and utilizing the nonlinear properties of glass, this radiation can be converted to a broad spectral distribution. The tight confinement puts limits on the maximum pulse fluence propagating in the core due to the damage threshold of the glass. Therefore, when higher power spectral densities are desired, it is favorable to spread the generation of light over a much larger area to prevent fiber damage. We present here a method for generating a supercontinuum using the cladding modes of an endlessly single-mode fiber. These modes generate a supercontinuum utilizing a multimodal quasi-continuum of states, for which dispersion is governed by the guiding properties of the material between the air-filled holes in the cladding. The system also provides experimental access to unique phenomena in nonlinear optics. Simulations of the propagation properties of the core mode and cladding modes were compared with measurements of the group-velocity dispersion in a modified white-light Mach–Zehnder interferometer. The coupling of similar laser parameters into the cladding of the photonic crystal fiber enables a significant increase in conversion efficiency in the visible spectral range compared with the core-pumped case.

**Keywords:** photonic crystal fiber; supercontinuum; cladding modes; endlessly single-mode fiber

## 1. Introduction

Supercontinuum (SC) generation is, especially for glass fibers, uniquely positioned among broadband light sources as it provides a very high degree of spatial coherence and, at the same time, high spectral power density, allowing light to be directed into areas that are normally difficult to illuminate [1–3]. Currently, several coherent, octave-spanning, high power SC sources have been realized using photonic crystal fibers (PCFs) [4,5]. In order to achieve broad spectral distributions with high power spectral densities, different technological approaches have been applied [6].

The rise in the ubiquity of these SC light sources circa 2010 [7] is due, in large part, to advances in the fabrication of specially designed optical fibers, known as PCFs. By tuning the properties of these PCFs, such as their core- and air-hole diameter, spacing and arrangement, the guiding properties of the fiber can be adapted for a myriad of applications such as a zero-dispersion fiber [8] or an endlessly single-mode fiber (ESF). An ESF contains only one core-guided mode over an extremely broad wavelength range [9]. In fact, it has been shown that the glass between the air holes in the cladding of an ESF can also guide light [10,11]. These modes, which are guided outside of the core, are known as ESF cladding modes. A study of these cladding modes [12] indicated that their dispersion characteristics are well suited to SC generation, while also maintaining a large cross-sectional area. These cladding modes provide coupling of radiation between individual cladding modes because of similarly tailored dispersion properties. Previous research has shown that, during supercontinuum generation in a polarization-maintaining PCF, the part of the pump laser light guided in the cladding contributes to generation of the supercontinuum [13]. The investigations carried out for the present work verified both experimentally and numerically that these ESF cladding modes had ideal dispersion characteristics for SC generation. Experiments conclusively demonstrated that it was feasible to couple enough power into these modes such that they did, in fact, efficiently generate an SC. Somewhat more interestingly, they did so with a higher efficiency in the visible spectrum than the standard core-pumped scenario.

## 2. Materials and Methods

### 2.1. Numerical Simulation of Fiber Parameters

The PCF (produced by fiberware, Mittweida, Germany) investigated here had a hole diameter of $d = 1 \pm 0.3$ μm and a hole-to-hole distance of $\Lambda = 3 \pm 0.2$ μm, which was measured with a MIRA 3 TESCAN scanning electron microscope (SEM). Similar fibers were shown to allow only single-mode propagation of light over wide spectral ranges in the core [9]. The fibers investigated here, however, deviated significantly from the theoretical, cleanly periodic structure of the PCF and to gain a greater understanding of the influence of this fact on the SC generation in these fibers, the optical modes of the PCF were numerically simulated using the following technique. Firstly, a grayscale image of the cross-section of the PCF was taken on the SEM and numerically converted into a binary image by setting a defined threshold value. In this way, the binary image numerically filtered noise, particles and scratches, producing a clean representative glass, no-glass image.

This binary image of the fiber's cross-section was used (taking the known index of refraction for glass and air) as the input to solve the vector Helmholtz equation. The fiber was made from high-purity synthetic quartz glass with low OH content (F300 Heraeus Quarzglas, Hanau, Germany). An numerical routine was implemented similar to the one found in Ref. [14], which was modified to also consider polarization. At a given wavelength, a sparse eigenvalue routine was used to solve the resulting system for both the fundamental mode, as well as the higher order cladding modes of the fiber. This routine provided a few of the spatial eigenfunctions, pictured in Figure 3b–f, and their corresponding propagation constants, $\beta(\lambda)$. For each mode, the group delay per unit length in Figure 4 was calculated based on this determined $\beta$ by:

$$T_g = \frac{1}{c}\left(n_{eff} - \lambda\frac{dn_{eff}}{d\lambda}\right),$$ (1)

where $n_{eff} = \beta/k_0$ is the effective index of refraction, $k_0$ is the free-space wavenumber, and $c$ is the speed of light.

### 2.2. Characterization of Group-Velocity Dispersion

The group-velocity dispersion (GVD) of different modes was experimentally measured using a Mach–Zehnder interferometer with a controllable delay line to investigate the time delay and find

the equalization wavelength for different modes (Figure 1) [12,15]. A camera at the free output of the second polarization-independent beam splitter allowed control over the near field pattern of the analyzed guided modes. An aperture in the image plane enabled the selection and suppression of individual modes. For the setup, a fiber-coupled SC light source, developed and manufactured in cooperation with fiberware GmbH, was used. A monochrome camera (acA2040-90umNIR, Basler, Ahrensburg, Germany), used during alignment, had an enhanced sensitivity in the near infrared spectral range up to 900 nm and a dynamic range of 59 dB. The test fiber was placed in a straight configuration with a length of 0.46 m ± 0.001 m in the interferometer.

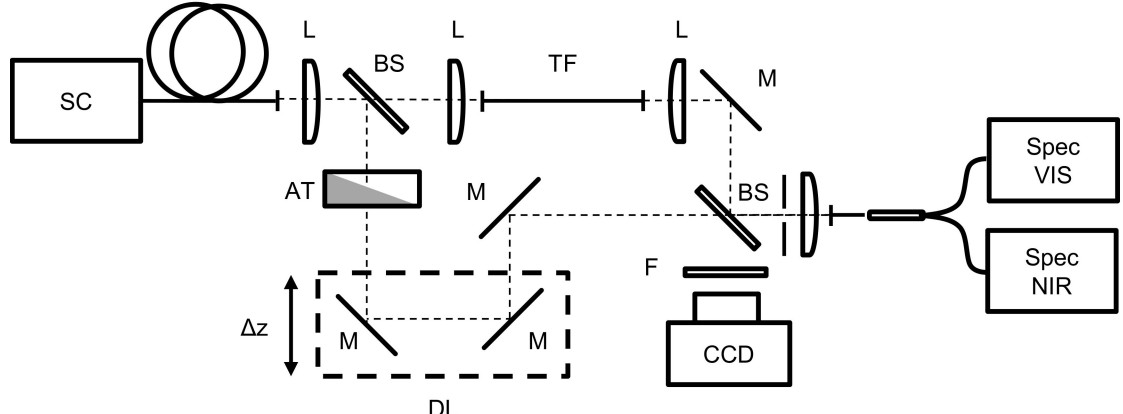

**Figure 1.** The interferometric setup used for measuring the group-velocity dispersion (GVD), similar to that used in Ref. [12]. SC, supercontinuum light source; BS, nonpolarizing beam splitter; DL, delay line; M, mirror; TF, test fiber; AT, attenuator; F, bandpass filter; L, lens; CCD, camera; and Spec, spectrometer.

## 2.3. Supercontinuum Generation

A particular challenge was efficiently coupling the high intensities of pump light necessary for SC into the cladding modes of the fiber. However, a tapered-fiber method was developed [16], which aided significantly in the process. Figure 2a shows a tapered section of the ESF which was created with the arc of a fusion splicer, while the fiber was fixed at each end but not under initial tension. The fiber was then cleaved at the center where the air holes had completely collapsed. At the in-coupling side, this taper formed a solid piece of glass which allowed the light to be pulled from a Gaussian focused, solid profile into the more complex mode-intensity profiles of the cladding modes. The light of a diode seeded picosecond fiber laser (FL) (IPG Laser, Burbach, Germany), with a pulse length of 180 ps operating at 1060 nm (up to 330 kW peak power and 22.5 W average power) was launched into to the cladding. A small numerical aperture lens was selected to match the taper angle of 2.99 degrees and the laser, $M^2 = 2$, was focused with a $1/e^2$ spot size of about 45 µm ($w_0 = 22.59$ µm, NA = 0.06 and a Rayleigh length of 187 µm) onto the collapsed part of the fiber. Figure 2b,c shows the output of the fiber at the pump wavelength after adjusting the system to couple into the core-guided mode and cladding modes.

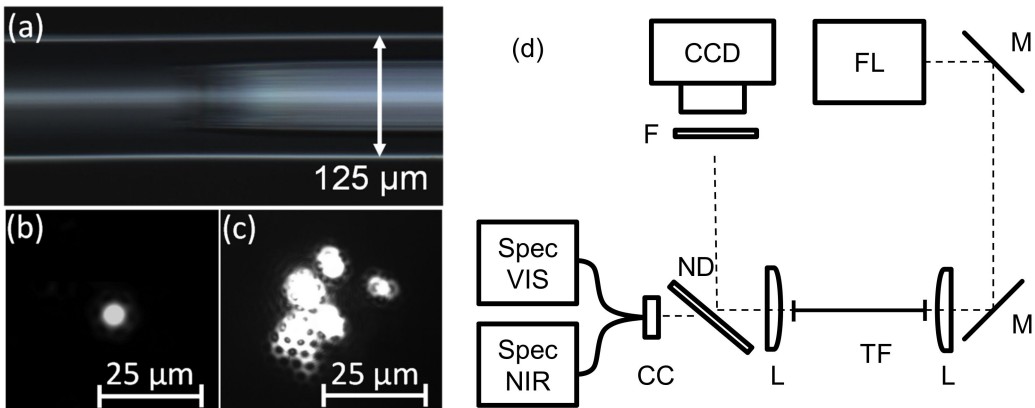

**Figure 2.** (**a**) Taper used to efficiently couple pump light into the microstructure, (**b**) core-coupled light spatial intensity pattern at 1060 nm and (**c**) cladding-coupled light intensity pattern at 1060 nm. Both images were captured on a logarithmic camera. (**d**) Shows the schematic of the optical setup with FL, fiber laser; M, mirror; L, lens; TF, test fiber; ND, reflective neutral density filter; CC, cosine corrector; F, bandpass filter; CCD, camera; and Spec, spectrometer.

When the light was coupled into the cladding of the PCF, some fiber modes showed a large loss after a few centimeters of propagation in the fiber. After a fiber length of 5 meters, 45.6% of the laser power was still measured, which corresponded to a coupling efficiency of 50.4% measured at 0.46 meters. The core-guided mode and fundamental cladding mode (seen in Figure 2c and calculated in Figure 3c) were the primary modes that were coupled into due to their spatially flat phase profiles and the fact that the pump light also had a flat phase profile at its focus. A schematic of the setup used to selectively launch the light into the core or cladding is shown in Figure 2d. The measurement of the spectral power densities was performed with two spectrometers (AvaSpec-ULS3648 and AvaSpec-NIR256-1.7 from Avantes) with a cosine corrector in a calibrated setup. For the investigation of the spatial intensity pattern in the core and cladding of the PCF at 1060 nm, the monochrome CCD camera (acA2040-90um, Basler, Ahrensburg, Germany) was applied (Figure 2b,c and at 850 nm in Figure 6e). An camera with color filter array (acA2040-90uc, Basler, Ahrensburg, Germany) was used to examine the SC at defined wavelengths in the visible (Figure 5 inset and Figure 6a–d) and a indium gallium arsenide camera (Bobcat-640-GigE, Xenics, Leuven, Belgium) was used in the infrared (Figure 6f–h).

## 3. Results

### 3.1. Linear Properties of the Endlessly Single-Mode Fiber

The single-mode propagation of light over a wide spectral range distinguishes ESF from conventional step index fibers. Besides the fundamental mode of an ESF, cladding modes can also occur and overlap spatially with the core. The core-guided mode is the only mode that is strongly localized and bound to the center of the fiber and very weakly interacts with the edge of the cladding-coating interface. The modes which overlap mostly with the photonic crystal region (the ones we refer to as cladding modes) require an additional reflection with the cladding–coating interface to achieve optical guidance. Despite that fact, they still have negligible propagation losses over distances of a few meters. The cladding ESF modes, as seen in Figure 3c–f, display similar symmetry to that of their multimode fiber counterparts [4], and are labeled with similar angular and radial quantum numbers. Numerical solutions to these modes push the boundaries of the finite element method and such calculations usually involve exploiting a 4-fold symmetry to aid in reducing the memory and time complexity required for the solution. This can be seen in how precise the effective index must be calculated. For the calculations performed in this manuscript, however, the 4-fold symmetry was relaxed. The modes seen in Figure 3c–f seemingly do not contain the proper mode-field distribution that their labels would suggest due to the somewhat random nature of the hole size and spacing in the

photonic crystal; however, the calculated phase profile was used as an indicator for mode classification. Based on the scanning electron microscopic image (see Figure 3a) the intensity distributions of the fundamental mode (FM) and the first cladding modes (CM$_i$) were simulated and shown in Figure 3b–f.

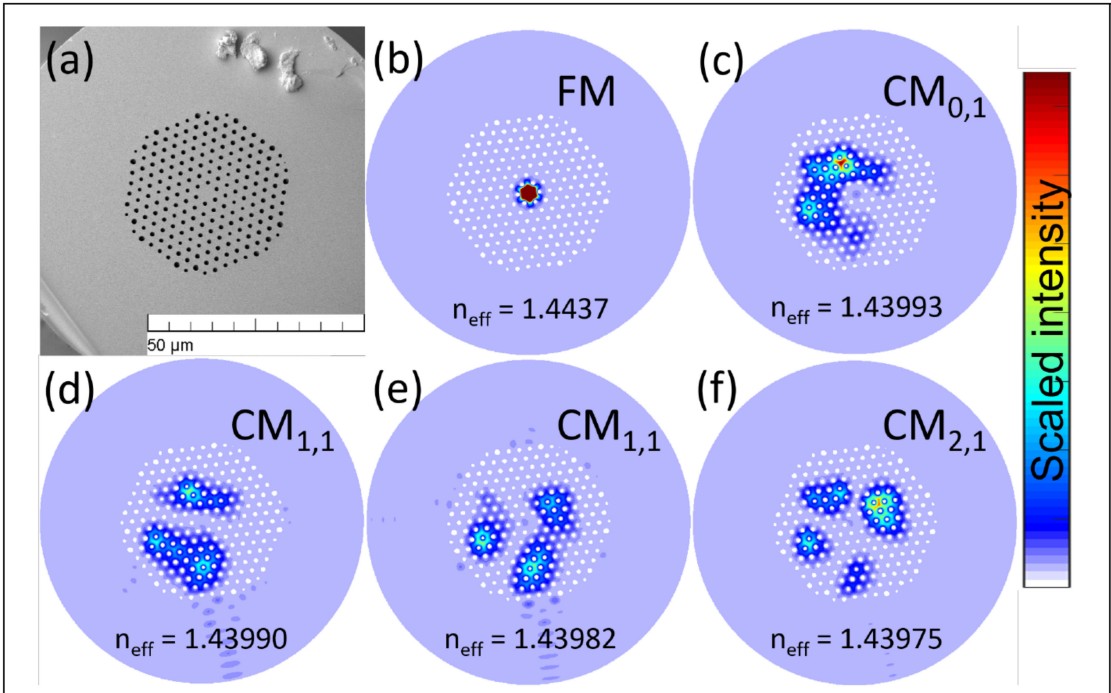

**Figure 3.** (**a**) Scanning electron microscope image of the endlessly single-mode fiber (ESF). The simulated ($\lambda$ = 1060 nm) modes of the ESF are as follows: (**b**) single core-guided mode and (**c–f**) a few higher-order cladding modes (CM), with the first subscript being the angular quantum number and the second subscript being the radial. The effective index of refraction n$_{eff}$ is shown for each mode.

### 3.2. Nonlinear Properties of the Endlessly Single-Mode Fiber

Nonlinear optical behavior between these cladding modes provides a different perspective to study four-wave mixing and soliton dynamics because there are a large number of optical modes spatially overlapping one another. The equations of motion governing such a system become intrinsically more complex [17,18] than those of their single-mode counterparts. Because of their large size yet relatively strong confinement, the dispersion characteristics of cladding modes are very nearly optimal for SC generation when pumped at 1060 nm. Also, due to their similar propagation constants, the power is easily coupled from one eigenmode to the next through relatively small, but non-zero perturbations to the equations of motion giving rise to their linear guiding properties. Four-wave mixing is the main process that serves to generate SC in the visible part of the spectrum. Although recent experiments in multimode gradient-index (MM-GRIN) fibers [19,20] and few mode fibers [21–23] have also realized SC generation in strongly overlapping states, the very different behavior of the propagating modes, as well as having their anomalous dispersion spectrally near the pump wavelength, changes the dynamics of how modulation instability and the Raman continuum [20] aid in formation of the spectral power density.

### 3.3. Dispersion Properties

One main advantage PCFs possess over step-index fibers when used for SC generation is that the dispersion characteristics of the PCF can be tuned by selecting the proper ratio of $\Lambda/d$ [3], where d is the cladding hole size and $\Lambda$ is the cladding hole spacing. A near-minimum value of the dispersion at the pump wavelength is desirable when generating SC [24–27]. Typically, the shorter wavelength

dispersion is dominated by the material dispersion and the longer wavelength dispersion is dictated by confinement. Under general conditions, if the mode area becomes larger, the region of anomalous dispersion is pushed towards longer wavelengths due to the weakened effects of confinement [28]. In the case of ESF, however, these large area modes of the cladding can still have a relatively short-wavelength zero dispersion because of the additional confinement provided by close air-hole spacing. These types of effects are also seen in supermode fibers, where the overall dispersion properties of the supermode are strongly governed by the properties of each individual core that make up the supermode [29]. This allows the intensity of a supermode to spread out over all of the coupled cores of the fiber while maintaining the dispersion characteristics of the individual fibers.

The group delay per length of the ESF was experimentally measured (Figure 4) using the interferometric setup with a spatial aperture added to the output of the fiber to filter particular modes. In Figure 4, two of the measured modes are labeled CMi;j because they are known to be cladding modes, but their exact quantum numbers are unknown. The modes' propagation constants were also calculated using the modified finite-difference algorithm introduced above. As is evident in the group-delay measurements in Figure 4, there are two distinct groups of curves. The lower curve represents the fundamental, core-guided mode, while the upper group of curves represent the dispersion of the cluster of cladding modes. In this case, the zero-dispersion wavelength for all modes falls in the region of 1060 nm, which is ideal for pumping with an ytterbium-doped fiber laser. Calculations showed that there were a very large number of modes associated with this upper group of cladding-mode curves, although it was not possible to experimentally distinguish all of them due to their complex spatial patterns.

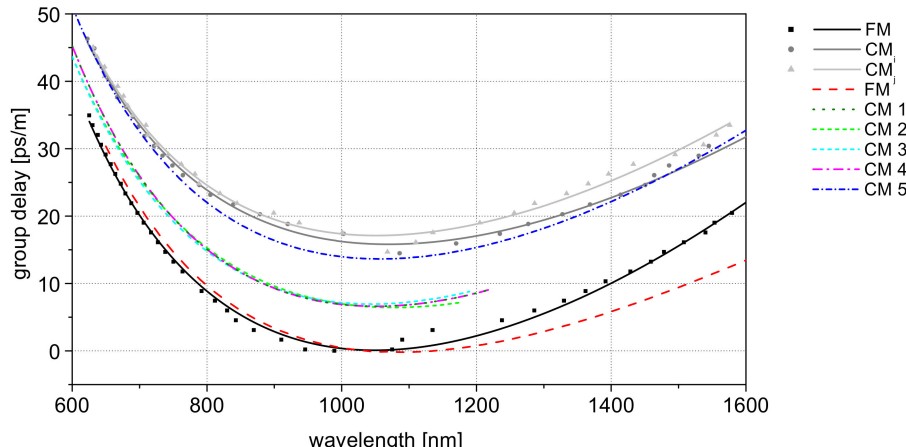

**Figure 4.** Measured (markers), fitted (solid) and calculated (dashed lines) curves of the group delay in the ESF for a 0.461 m length of the fiber. FM, fundamental mode and CMi, arbitrarily labeled cladding mode.

### 3.4. Supercontinuum in the Core and Cladding

To demonstrate the differences in single-mode SC generation from that in the multimode generation, as well as show that an SC could be generated in the cladding modes (as opposed to the scenario where the continuum was generated in the core and scattered from the core-guided mode to the cladding), the pump light was first coupled into the core-guided mode. The resulting SC spatial pattern of the output can be observed in Figure 5 (inset). As is visible in the spectrally integrated pattern, most of the light generated inside the core-pumped case remained there. The red, dashed curve in Figure 5 shows the spectral power density of the core-guided SC generation when pumped just below the damage threshold for that configuration. The power limitation of this arrangement, as well as the cladding-pump arrangement, was determined by the coupling losses heating the end of the fiber, which would cause enough thermal expansion inside of the ceramic holder to fracture the fiber. It can be seen that the core-pumped intensity in the Raman continuum (generated in the

anomalous dispersion regime at wavelengths longer than the pump) is almost one order of magnitude greater than the SC generated on the shorter wavelength side of the pump.

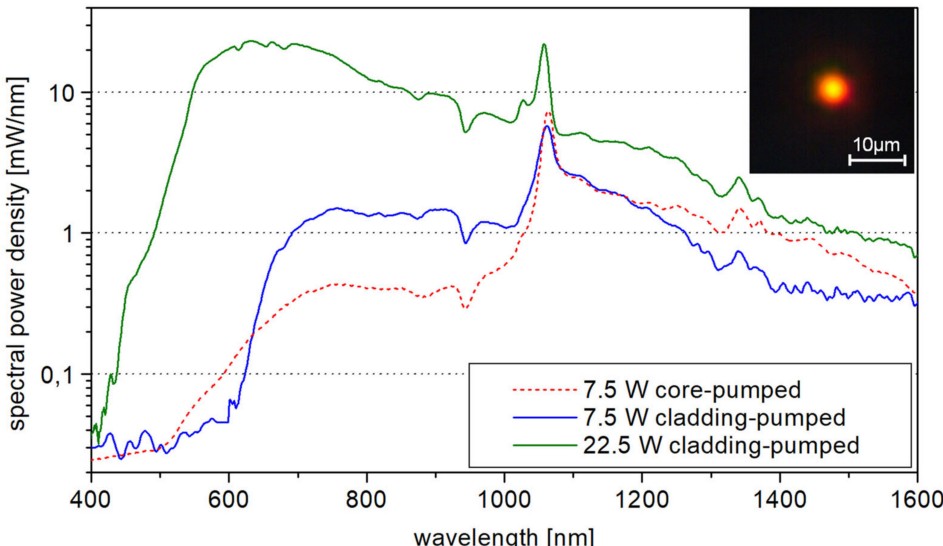

**Figure 5.** Output spectral power density from 5 m of ESF for the core-pumped SC (red solid curve) and cladding-pumped SC (blue dashed curve) when pumped with 7.5 W. Highest achievable SC output in the cladding modes with 22.5 W pump (green). Core-pumped spectrally integrated profile (inset). The pump was held at a constant 60 kHz repetition rate.

Figure 6a shows the spectrally integrated results of the spatially resolved SC output (under the same pumping conditions as Figures 2c and 5's green curve) after propagating to the end of a 5 m long piece of fiber. It is very interesting to note the encircled area of Figure 6a, which shows a location in the cladding which appears to strongly favor SC generation. Even under adjustment of the pump conditions, and therefore the superposition of cladding modes, the pump was coupled into this position and the fiber always demonstrated a high propensity for the generation of light in the visible spectrum.

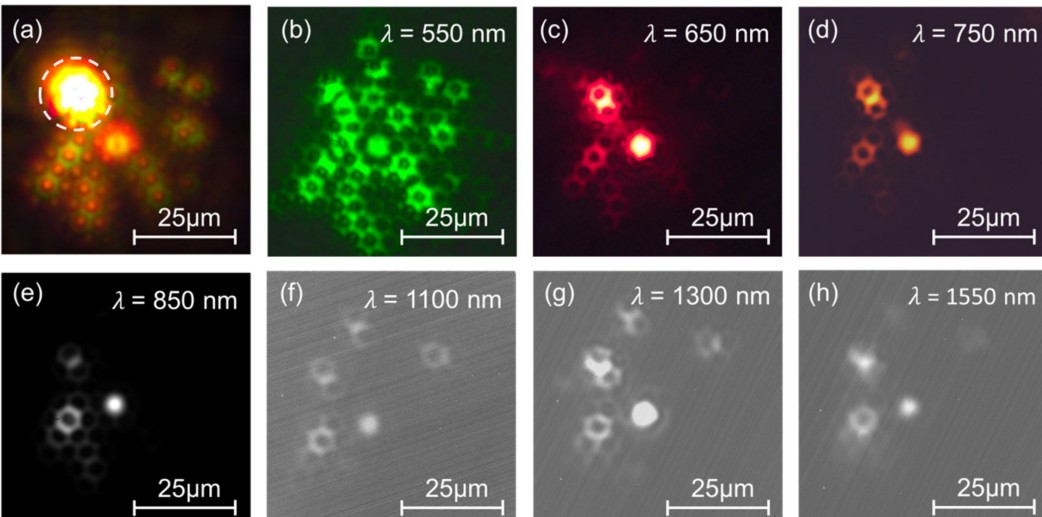

**Figure 6.** The spatial profile of the output of SC generated inside the core: (**a**) Multispectral and filtered for the marked wavelengths (**b**–**h**). The encircled area of the picture in (**a**) shows a place where the air-hole glass size and spacing strongly enhance SC generation.

## 4. Discussion

In the case for the cladding-pumped SC generation, the pump light was coupled into a superposition of the core-guided mode and the fundamental cladding mode of the fiber since the input coupling mechanisms did not allow much freedom to select other desired modes. The selection of other superpositions was difficult to achieve and the resulting spatial output pattern of the pump was very stable over the course of the measurements once the pump's launching conditions were set. It is believed that the lowest-order cladding mode, shown in Figure 3c, was mostly excited due to the comparable intensity profile of the measured mode with the calculated mode. The $CM_{0,1}$ mode also maintained a flat phase profile that would match that of the pump laser near its focus. Both the spatial position of the pump, as well as the pump angle relative to the face of the fiber, were adjusted to produce the widest spectrum on the output. The blue, solid curve in Figure 5 shows the power spectral density for the same pump powers as in the core-pumped damage-limited case. Notice that the SC generation spectrum does not extend as far into the visible in the cladding-pumped case vs the core-pumped case under similar pump powers because the spatial mode size for the cladding-pumped mode is larger and the nonlinearity is therefore lower. However, due to the increased pump area in the cladding mode scenario, much higher pump powers were achieved before the fiber was damaged and a much brighter SC was produced (green solid curve in Figure 5). The spectral power density in the high-power cladding-pumped case showed a much larger SC generation in the visible regime most likely due to the large number of modes which are phase matched to the pump mode. The dynamics of SC generation did not follow the pump in the spatial pattern but they were influenced strongly by it. This is probably an effect of four-wave mixing, which allows for intermodal mixing at certain wavelengths if the phase velocity of the modes is similar; however, this is only speculation at the moment. Considering Figure 6c–d, the generated light follows the intensity distribution of the pump light, shown in Figure 2c, but when the generated light reaches shorter wavelengths, the spatial pattern becomes more spatially dispersed and at 550 nm there seems to be a transition to a more chaotic mode pattern.

## 5. Conclusions

Broadband SC was realized utilizing the cladding modes of an endlessly single-mode photonic crystal fiber. The modes had sufficient dispersion and, using a tapered-fiber optical coupling technique, enough power was coupled into the cladding to generate an SC with a higher spectral power density than could be achieved when only core-pumping the fiber. The cladding modes were shown, both interferometrically and numerically, to be near the optimum zero-dispersion wavelength for SC generation when pumped with a 1060 nm laser. Tapered-fiber preparation allowed for the efficient coupling of laser radiation into the cladding modes (up to 50.4% efficiency). This was sufficient to surpass the core-pumped SC generation in overall output power before the fiber was damaged. The conversion of laser radiation in a multimode waveguide with adapted dispersion was able to overcome the maximum power limit set by the damage threshold in core-pumped SC, but this came at a cost of reduced spatial coherence in the final spectrally-integrated SC output. The nonlinear dynamics of light in a multimode configuration were similar to, but different from, the previous experiments presented in an MM-GRIN fiber [20] and will provide an interesting platform for exploration owing to the fact that the light stays approximately in the same spatial superposition it was launched into or generated in. The modes can still be dispersed with relatively weak perturbations for the visible spectral range, such as when bending of the fiber occurs. Furthermore, there appear to be defects in the air-hole cladding which generate SC more readily than others. The core is one of these defects, but there are other spatial locations too. Lastly, the spectral power densities realized in the visible spectral range are among the highest achieved so far. Since no damage of the fiber ends occurred, even at maximum pumping power, further power scaling is possible in the future. It is believed this high-efficiency conversion into the visible is due to a large number of modes which are optimum for

4-wave mixing with the pump, which made the conversion process much more efficient. However, the mechanisms supporting such an idea are yet to be confirmed.

**Author Contributions:** T.B. carried out the dispersion measurements of the fiber and the investigation of the spectral conversion of the laser radiation in the core and cladding. B.N. simulated the propagation properties of the fiber numerically. Together they developed the method and wrote the paper with the scientific support of A.F.L. and P.H.

**Funding:** The authors would like to thank the Saxon State Ministry of Science and Art for financial support under grant number 4-7544.10/7/3.

**Acknowledgments:** The authors would like to thank fiberware, Inc. for providing us with the endlessly single-mode fiber and members of the Optical Technologies working group at the West Saxon University of Applied Sciences and members of the Fraunhofer Institute for Material and Beam Technologies at the Application Center for Optical Metrology and Surface Technologies for their fruitful discussions and preparatory lab work.

**Conflicts of Interest:** The authors declare no conflicts of interest.

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
