# Peer review of "Supercontinuum Generation in the Cladding Modes of an Endlessly Single-Mode Fiber"

_applsci, doi:10.3390/app9204428_

Round 1

Reviewer 1 Report

Supercontinuum generation (SG) was realized using cladding modes in a tapered single-mode photonic crystal fiber (PCF). By couple pumper laser to both core and cladding region, the effective area for SG is increased and higher pump power can be used for the SG. The SG stimulated from both core and cladding modes was demonstrated.

The idea of using cladding modes for SG is interesting. The reviewer has the following questions and suggestions need to be addressed before this paper can be published.

The reviewer did not see any reference related to the SG using cladding modes in this manuscript. Is this manuscript the first one that studies SG using cladding modes? If not, the reviewer suggests the authors to include the pioneering works from others. If not, the reviewer would like to know why this kind of study has not been studied by others yet since the SG in PCF has been studied for ~20 years. The authors mentioned, “Despite that fact, they still have negligible propagation losses over distances of a few meters.” The reviewer would like to know the loss of cladding modes in this PCF and the length of the fiber that was used in the experiment. Figure 3 is mentioned earlier than Fig. 2 in the manuscript. The reviewer suggests that the order of Fig. 2 and Fig. 3 should be switched In Fig. 5, is 22.5 W the threshold of the cladding pumped-method?

Author Response

Dear Editors,

We are grateful for the reviewers’ feedback given to improve this manuscript and we are also grateful for the opportunity to resubmit the document with revisions. A list of complete changes that were made to the manuscript is given below, including those suggested by the reviewers and those necessary to meet the proper journal formatting.

The reviewer did not see any reference related to the SG using cladding modes in this manuscript. Is this manuscript the first one that studies SG using cladding modes? If not, the reviewer suggests the authors to include the pioneering works from others. If not, the reviewer would like to know why this kind of study has not been studied by others yet since the SG in PCF has been studied for ~20 years.

The authors have done an extensive literature search, and as far as we can tell, could not find an instance where the cladding modes of a PCF were used to generate a supercontinuum. Part of the reason we believe this has not been observed was the challenge of efficiently coupling light into these modes as well as the very computationally expensive calculations necessary to realize accurate dispersion curves for these modes. In fact, some references given in the submitted manuscript for the calculation of and analysis of PCF cladding modes (Refs. 10 and 13), were done as recently as 2016. In the beginning of 2019, a publication references the possibility that the cladding modes may participate in supercontinuum generation (“Investigation of supercontinuum generated in the cladding of highly nonlinear photonic crystal fiber”, Vengelis, et. al) but never fully took advantage of the cladding modes. We have added the reference 29 in the introduction.

The authors mentioned, “Despite that fact, they still have negligible propagation losses over distances of a few meters.” The reviewer would like to know the loss of cladding modes in this PCF and the length of the fiber that was used in the experiment.

To address this point, the following text was included to the methods section of the paper, ” When the light is coupled into the cladding of the PCF, some fiber modes show a large loss after a few centimeters of propagation in the fiber. After a fiber length of 5 meters, 45.6% of the laser power can still be measured which corresponds to the coupling efficiency of 50.4%.”

The light in the cladding modes was still visible after 150 m. Corresponding investigations were carried out with very low laser power at a wavelength of 1060 nm.

Figure 3 is mentioned earlier than Fig. 2 in the manuscript. The reviewer suggests that the order of Fig. 2 and Fig. 3 should be switched In Fig. 5, is 22.5 W the threshold of the cladding pumped-method?

We have changed the incorrect numbering and marked the corresponding changes in blue in the manuscript. The incorrect laser power of 20 W was also changed to the correct value of 22.5 W and marked.

Reviewer 2 Report

For comments see attached file

Author Response

Dear Editors,

We are grateful for the reviewers’ feedback given to improve this manuscript and we are also grateful for the opportunity to resubmit the document with revisions. A list of complete changes that were made to the manuscript is given below, including those suggested by the reviewers and those necessary to meet the proper journal formatting.

The paper contains a lot of abbreviations, but very often abbreviations are not explaining in the right place. I recommend PCF and SC abbreviations explain in the introduction chapter. GWD abbreviation is mention in the abstract part but I recommend adding an explanation in chapter 2.2. Characterization of dispersion and also on the label for Fig. 1.

We have changed the description of the acronyms as suggested and marked the corresponding changes in blue in the manuscript.

In chapter 2.1. Numerical simulation of fiber parameters, I do not understand what simulations were done, how they were done (what methods were used) and where the simulations are presented in the paper? If I understand well, the authors used commercially available PCF, I recommended providing more details about this PCF.

A reference was given [28] to follow the method used in the simulations. We have added further information on the PCF used in chapter 2.1 and marked it blue. It was a noncommercial fiber produced for us by fiberware, Inc.

In chapter 2.2. Characterization of dispersion / describing Fig. 1 in more detail

We have added the description of the setup and marked the changes in blue.

For Figs. 2(a), (b) and (c) are not clear how were these Figs. obtained and I recommend add scale for Figs. 2(b) and (c), delete the comment λ= 1060 nm and add this information on the label for this Fig. Fig. 2(d) mention that the authors used FL - fiber laser. It is necessary to provide technical information about this FL and I also expect comments if the Spec. presented in Fig. 1 are the same as Fig. 2.

We have changed the incorrect numbering and marked the corresponding changes in blue in the manuscript. In order to supplement the missing information about the fiber laser used, we have added the following text: “The cladding was then trivially, although not highly efficiently, coupled in to with a diode seeded picosecond fiber laser - FL (IPG Photonics), with a pulse length of 180 ps operating at 1060 nm (up to 330 kW peak power and 22.5 W average power). The measurement of the spectral power densities was performed with two spectrometers (AvaSpec-ULS3648 and vaSpec-NIR256-1.7 from Avantes) with a cosine corrector in a calibrated setup.”

Fig. 3(a) shows a picture obtained by scanning electron microscope (SEM) image, but the description of this SEM is missing in chapter 2 Materials and Methods?

We have added the description of the fiber, the scanning electron microscope and a link to figure 3 (a) in chapter 2.1.

I would like to also ask you if the results presented in Figs. 3(b)-(f) were measured or it is the simulation? The description for these Figs. 3 is not sufficient and it is not the clear difference between Fig. 3(d) and Fig. 3(e)?

We have supplemented the description of the simulated intensity distributions and described the shortcuts used. The modes were simulated based on the data extracted from the SEM and the method referenced in [28].

I would like to ask you why is the difference measurement range for wavelength presented for Fig. 4 (λ= 600 – 1500 nm) and for Fig. 5 (λ= 350 – 1600 nm)? I would like to ask you what does it mean FSM and CM1-5 on the label for Fig. 4? I would like to also recommend to rewrite comments for the inset picture in Fig. 5 and provide the scale for this figure.

We have adapted the diagrams as suggested and have complemented the lettering and scale. The scale in Fig. 4 was set by the bounds of the computations but has been updated per your request. FSM has been changed to FM to reflect previous usage and due to the method used to measure dispersion, an exact radial and angular label could not be given to the cladding mode. The caption has been updated to reflect this.

Fig. 5 (row 173) presented output spectral power density for 5 m of ESF for the core-pumped SC and cladding-pumped SC. On label for this Fig. are mention power 7.5 W core-pumped and 7.5 or 22.5 W cladding-pumped. Therefore, I do not understand the statement on row 175 - Highest achievable SC output in the cladding modes with 20 W pump.

We have changed the incorrect numbering and marked the corresponding changes in blue in the manuscript. The incorrect laser power of 20 W was also changed to the correct value of 22.5 W and marked.

I would like to ask the authors why is Fig. 5 row 201 presented in the conclusion chapter and why the authors chose presented picture for filtered wavelength for 550 nm, 650 nm, and 750 nm?

Fig. 6 was moved to its intended location in the results section. The guiding properties of the PCF show clear wavelength dependence and at shorter wavelengths the entire cladding guides light. Comparing Fig. 6(c) to 6(d) the generated light follows the intensity distribution of the pump light, Fig. 2(c), but when the generated light reaches shorter wavelengths, the spatial pattern becomes more spatially dispersed. We chose these wavelengths to show that the generated continuum at 750 nm and 650 nm have the same mode profile, but at 550 nm there seems to be a transition to a more chaotic mode pattern. The authors found this point interesting although still have no explanation or speculation as to why this may be.

English correction

We have made thorough corrections to the English. Any changes are highlighted in the revised manuscript.

Round 2

Reviewer 2 Report

for comments see attached file

Author Response

Dear reviewer,

We are grateful for your feedback on improving this manuscript and would like to apologize for the incomplete editing of the comments. We are also grateful for the opportunity to resubmit the document with major corrections. Below is a list of the changes made to the manuscript.

Reviewer 2

In the first revision, I suggested providing explanation abbreviations in the Introduction chapter. In this chapter, are still are abbreviations PCF and SC without definition. I understand that PCF is mention in caption Featured application and CS in the abstract part but I recommended do not use abbreviations in abstract and provide this information in the later text.

Our apologies, we mistakenly understood the remarks during the first editing and have changed the description of the acronyms and marked the corresponding changes in green in the manuscript.

The authors add in the introduction chapter new cite literature 29. I recommend renumbering cite literature. For this new cite paper also missing information about the names of the authors.

We renumbered the citations and added the missing information.

In my first comments, I mention that in chapter 2.1. Numerical simulation of fiber parameters, I do not understand what simulations were done, how they were done (what methods were used) and where the simulations are presented in the paper? What does it mean statement: The optical properties of the PCF were numerically simulated. I think it necessary to provide information about the methods which were used for simulation in more details?

We have added the description about the methods which were used for simulation in more details. We hope the manuscript states more clearly that the modes and propagation constants were numerically calculated using the scanning electron microscope of the fiber which were used in a numerical eigenvector routine. The calculated wavelength-dependent propagation constants were then used to determine the group delay.

I had also remarks for chapter Characterization of Group Velocity Dispersion (GVD). The most required informations were added but I would like to ask to provide also the description for the used CCD camera. The presented pictures on Fig. 2b, c provide the pictures of the light with wavelength 1030 nm but common CCD cameras are used for visible spectrum?

We added the information about the cameras and where each was used in the figures.

On Fig. 2b due to my previous request was add scale but the scale is too large 25 μm. It is difficult to know if the diameter of the light spot on the picture is 3 μm or more? I would like to ask you if the picture of the light beam on Fig. 2b is the same as on Fig. 5?

We corrected the scale and the image in Fig. 2b, and added text stating the image is recorded at the pump laser wavelength of 1060 nm. The inset in Fig. 5 is different than 2(b) (visible light captured with RGB camera).

I would like to ask you to comment length of the test fiber in more details. The authors provide information on page 3: After a fiber length of 5 meters, 45.6% of the laser power can 90 still be measured which corresponds to the coupling efficiency of 50.4%. The authors tested the different length of the PCF in the setup on Figs. 1, 2?

Yes, we tested the different length of the PCF in the setup 2, replaced the spectrometer with a power meter and operated the laser at very low power to eliminate nonlinear effects. The fiber length to determine the coupling efficiency has been added in the manuscript.

On label for Fig. 3 there are twice mention for Fig. (a). The description for results also presented in Figs 3(b)-(f) are insufficient and there is not clear how were these results obtained? In my opinion, the statement that the applied method is in reference [28] is not adequate. It is not clear why on Fig. 3(d) CM1,1 is neff= 1.43990 and on Fig. 3(e) CM1,1 is neff= 1.43982. I would like to ask you if it is possible to compare simulated results with real measurement results?

We added the Information of the method. The difference of both cladding modes were caused by structural inhomogeneities of the microstructure. The intent of the labeling CM1,1 was to say that this mode exhibited similar properties to the LP1,1 mode. However, in this case, as the symmetry of the standard multimode fiber is lifted by introducing the randomness of the air holes’ size and shape, the degeneracy of the CM1,1 state is also lifted. We therefore find different propagation constants for the two spatial distributions of the CM1,1 mode as well as different effective indices of refraction. Unfortunately, it is very hard to study these modes in the laboratory because they are hard to distinguish from one another and an exact comparison between theory and experiment is challenging.

I would like to ask you why the calculated data shown in Fig. 4 for CM2 and CM4 are provide only for wavelength 600-1210 nm. The data for all other results are provided for the whole range up to 1600 nm. Where is data for CM3? It seems to me that curve for CM3 is provided as the white line and it is not possible to read it on white background.

The calculated data shown in Fig. 4 for CM2 and CM4 are provided only for wavelength 600-1210 nm because the corresponding modes cannot be guided at higher wavelengths. We changed the line with of CM3 in the plot.

In Fig. 6 are missing scales. I had also previous comments why authors chose presented picture for filtered wavelength for 550 nm, 650 nm, and 750 nm? The authors response was We chose these wavelengths to show that the generated continuum at 750 nm and 650 nm have the same mode profile, but at 550 nm there seems to be a transition to a more chaotic mode pattern. The authors found this point interesting although still have no explanation or speculation as to why this may be. The presented result is interesting but wavelength range for SC is from 400 to 1600 nm. It will be interesting to provide results for filters with higher wavelengths - for example, 850, 1300 and 1550 nm.

We provide results for filters with higher wavelengths 850 nm, 1100 nm, 1300 nm and 1550 nm.
